# Quantum reservoir probing: An inverse paradigm of quantum reservoir computing for exploring quantum many-body physics

Kaito Kobayashi and Yukitoshi Motome

Department of Applied Physics, the University of Tokyo, Bunkyo-ku, Tokyo 113-8656, Japan

## Abstract

Quantum reservoir computing (QRC) is a brain-inspired computational paradigm that exploits the natural dynamics of a quantum system for information processing. To date, a multitude of quantum systems have been utilized in QRC, with diverse computational capabilities demonstrated accordingly. This study proposes a reciprocal research direction: Probing quantum systems themselves through their information processing performance in the QRC framework. Building upon this concept, here we develop quantum reservoir probing (QRP), an inverse extension of the QRC. The QRP establishes an operator-level linkage between physical properties and computational performance. A systematic scan of this correspondence reveals the intrinsic quantum dynamics of the reservoir system from both computational and informational perspectives. Unifying quantum information and quantum matter, the QRP holds great promise as a potent tool for exploring various aspects of quantum many-body physics. In this study, we specifically apply it to analyze information propagation in a one-dimensional quantum Ising chain. We demonstrate that the QRP not only distinguishes between ballistic and diffusive information propagation, reflecting the system's dynamical characteristics, but also identifies system-specific information propagation channels, a distinct advantage over conventional methods.

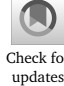

## 1   Introduction

The contemporary era has witnessed an extraordinary escalation in the capabilities of artificial intelligence. Emulating the intricate workings of the human brain, it has revolutionized diverse domains, including image recognition and machine translation [1–3]. Nevertheless, no matter how smart it is, artificial intelligence remains constrained by the fundamental physical limitations inherent in the silicon-based substrates on which it is realized. Considering the substantial energy consumption and the approaching downscaling limits, the need for alternative computational paradigms has been widely recognized. Consequently, unconventional computing now stands as an interdisciplinary frontier in scientific exploration [4–6]. A leading methodology in this domain is physical reservoir computing [7–12]. In this brain-inspired algorithm, an input-driven dynamical system, termed a physical reservoir, performs nonlinear transformations on sequential input data. When the dynamics exhibits a high-dimensional internal space with pronounced nonlinearity, a simple linear transformation of read-out results from the physical reservoir is sufficient to precisely generate a target output function. Quantum systems inherently satisfy these criteria, possessing intrinsic nonlinearity and an exponentially large Hilbert space. This has led to the development of quantum reservoir computing (QRC) framework, which leverages quantum systems as physical reservoirs [13,14]. Seminal proposals with spin-based implementations have demonstrated the exceptional performance of the QRC [13–18], later expanded to a variety of quantum systems including fermionic and bosonic networks [19–21], harmonic and nonlinear oscillators [22–25], and Rydberg atoms [26]. Furthermore, recent advancements in quantum technologies have facilitated proof-of-principle experiments for the QRC across several quantum reservoir settings, such as nuclear magnetic resonance systems [27] and superconducting qubits [28,29]. Importantly, the diverse computational capabilities observed across different types of QRC systems present an intriguing research avenue: The investigation of quantum systems through their computational performance when utilized within the QRC.

In this study, we propose an inverse extension of the QRC framework, termed quantum reservoir probing (QRP). While the QRC aims to exploit quantum systems as a reservoir for computational purposes, the QRP is specifically dedicated to elucidating quantum many-body physics of the reservoir from a computational point of view. Notably, in recent years, the intersection of quantum information and quantum matter has gained prominence in various contexts, highlighting the utility of quantum information in probing quantum many-body phenomena such as quantum chaos [30–36], thermalization dynamics [37–43], and dynamical quantum phase transitions [44–47]. Analogously, the QRP investigates quantum phenomena through an interdisciplinary approach by establishing a correspondence between the computational performance and the physical attributes of the employed quantum system. Since a variety of phenomena can be associated with computation by judiciously selecting information processed or computational tasks performed, the QRP has broad applications in the exploration of quantum many-body physics. As a fundamental demonstration of the research avenues via the QRP, we here investigate the dynamics of information propagation within quan-

tum systems, where locally encoded quantum information spreads over a multitude of degrees of freedom. Although such local information often becomes inaccessible to local probes as the dynamics progress toward the long-time limit (quantum information scrambling [48,49]), our study focuses on the early timescale, far from being fully scrambled, where understanding how information is distributed in the Hilbert space at each moment becomes a pertinent question.

In this application of the QRP, information is directly monitored analogously to a pump-probe paradigm. Random information is locally injected into the quantum system under investigation, and the system's response is recorded in a selected degree of freedom. Subsequently, the original input value is estimated using the observation outcomes based on a statistical approach. Successful estimation signifies that information has propagated to that read-out degree of freedom; otherwise, it remains unpropagated therein. Utilizing this estimation performance as an indicator, the QRP can comprehensively assess information propagation to an arbitrary degree of freedom at an arbitrary time in a unified manner. We demonstrate the efficacy of the QRP by investigating a one-dimensional quantum Ising chain as a paradigmatic example. We show that the QRP distinctly captures the information propagation dynamics that reflects the intrinsic dynamical characteristics of the system, such as quasiparticle-mediated propagation in an integrable free fermion system and correlation-mediated propagation in a quantum chaotic system. Moreover, by systematically scanning the read-out degrees of freedom, the QRP reveals the mechanisms governing information propagation between different degrees of freedom, namely information propagation channels, which are typically inaccessible via conventional methodologies. We believe that our QRP presents an interdisciplinary paradigm to further advance the understanding of quantum many-body physics.

## 2 Scheme of QRP

### 2.1 Concept of QRP and its relationship to QRC

Prior to exploring the QRP, we introduce the QRC, a computational paradigm specifically designed to leverage quantum systems for information processing [13]. The architecture of QRC is illustrated in Fig. 1(a), comprising three layers: Input, reservoir, and output. In the input layer, time-series input data is encoded onto a quantum system, specifically referred to as a quantum reservoir. The principal role of the reservoir layer is to nonlinearly project the input data into an internal feature space, effectively emulating a network of artificial neurons with recurrent pathways. Unlike conventional machine learning paradigms involving optimizations, the internal attributes of the quantum reservoir remain fixed, as predetermined by its inherent physical characteristics. This is analogous to leaving parameters within a neural network untrained, which leads to a substantial reduction in processing costs compared to schemes requiring the training of the entire weights. In the reservoir layer, the dynamics of the quantum reservoir system in response to inputs are recorded through measurements of specific variables. These read-out outcomes are accumulated into a one-dimensional state vector, which is then linearly transformed using a weight vector in the output layer. Only this weight is trained to produce the desired output for a given machine learning task. By leveraging the pronounced nonlinearity and high-dimensional Hilbert space of the quantum reservoir, the QRC can achieve robust neuromorphic computation solely through such simple linear post-processing.

To harness the full potential of the quantum reservoir and access the wealth of information encoded in the Hilbert space, a straightforward approach involves measuring multiple operators, thereby increasing the number of computational nodes in the post-processing stage. For example, when utilizing an $N$-site spin system as the quantum reservoir, single-site Pauli

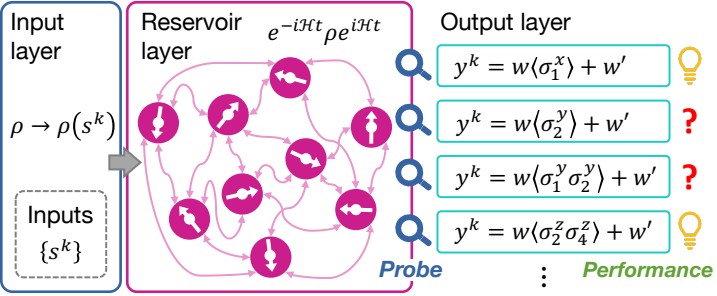

Figure 1: (a) Concept of the QRC. Sequential inputs $\{s^k\}$ are provided at the input layer, and the internal state of the quantum reservoir $\boldsymbol{X}^k$ is extracted based on measurements of various degrees of freedom. At the output layer, the final output $y^k$ is computed by linearly transforming $\boldsymbol{X}^k$ using the weight vector $\boldsymbol{w}$. (b) Schematic representation of the QRP. The final output is calculated using an individual degree of freedom, whose performance elucidates the internal structure of the Hilbert space.

measurements $\langle \sigma_i^\alpha \rangle$ ($1 \le i \le N, \alpha = x, y, z$) yield a set of $3N$ values, and two-site Pauli measurements $\langle \sigma_i^\alpha \sigma_j^\alpha \rangle$ ($1 \le i, j \le N, i \ne j$) generate a set of $3N(N-1)/2$ values. Combining these measurement results, a $(3N(N+1)/2 + 1)$-dimensional state vector is constructed, incorporating an additional constant term. The inclusion of read-out outcomes from longer Pauli strings appears to enhance computational performance. However, empirical evidence suggests that adequate performance can be achieved using only a number of degrees of freedom that scales polynomially with respect to $N$, beyond which further improvement tends to saturate [17,18,21]. Considering the exponentially large dimensionality of the Hilbert space, this implies that certain degrees of freedom may not effectively contribute to computation or may extract redundant information. Although the selection of read-out operators is often overlooked, the suitability of a particular degree of freedom for computation should reflect the intrinsic characteristics of the Hilbert space, providing insights into the underlying physics of the quantum reservoir system.

The QRP is the conceptual inverse of the QRC, diverging in their primary focus: While the QRC is oriented toward computational applications, the QRP is focused on uncovering underlying physical insights. This paradigm aligns with the recent unification of quantum information and quantum matter research, where quantum informational metrics are leveraged to unveil a variety of quantum phenomena. The QRP further accelerates this integration, being designed to illuminate quantum many-body physics through the computational capabilities of the quantum reservoir system. This work demonstrates the effectiveness of the QRP in analyzing information propagation, offering an alternative to established approaches that rely on, for example, many-body correlations, entanglement entropy, or mutual information.

Figure 1(b) schematically illustrates the architecture of the QRP. In contrast to the QRC, our QRP enhances resolution in accessing the Hilbert space by deliberately constraining the read-out to a single degree of freedom. In this framework, input is supplied to and transformed within the quantum reservoir system (similar to the QRC), and the final output is calculated from the measurement outcome of a single operator (different from the QRC). Under this condition, the computational performance is linked to the physical attributes of the observed degree of freedom. Specifically, we focus on the estimation task for the input value, the performance of which quantifies the memory of input information retained in the observed degree of freedom. Successful (unsuccessful) estimation indicates that the input information does (does not) influence the read-out operator, thus revealing whether the information has propagated to that degree of freedom. For example, in Fig. 1(b), the output derived from $\langle \sigma_1^x \rangle$ exhibits superior estimation performance, suggesting that information reaches $\sigma_1^x$ in the Hilbert space; conversely, the inferior performance obtained from $\langle \sigma_2^y \rangle$ indicates the information does not propagate to $\sigma_2^y$ possibly due to system-specific dynamics. Analogous analysis can be applied to any degree of freedom. Therefore, by systematically scanning read-out operators, the QRP enables the evaluation of information propagation in the Hilbert space with operator-level resolution. We note that the estimation task is meticulously selected for this research objective: By addressing alternative machine learning tasks, the QRP can probe diverse physical properties beyond information propagation.

## 2.2 QRP formulation to analyze information propagation

Let us formulate the QRP framework for the analysis of information propagation. Although we take a spin system as an illustrative example [Fig. 2(a)], we emphasize that the QRP framework is versatile and applicable to a variety of systems. We begin by introducing the input protocol, where we sequentially provide random information through local quantum quenches; the QRP also accommodates alternative input methods, such as input-dependent magnetic fields or electric currents. Suppose $\{s^k\}$ represents an input sequence with each $s^k$ randomly sampled from a uniform distribution: $s^k \in [0, 1]$. At every time interval $t_{\text{in}}$, the density matrix $\rho$ is updated as

$$\rho(kt_{\text{in}}) \rightarrow |\psi_{\text{in}}(s^k)\rangle\langle\psi_{\text{in}}(s^k)| \otimes \text{Tr}'[\rho(kt_{\text{in}})], \tag{1}$$

where $|\psi_{\text{in}}(s^k)\rangle$ represents the input state of the qubits used for encoding $s^k$, and $\text{Tr}'$ denotes the partial trace performed over these input qubits. Starting from the ground state, a total of $l^{\text{w}} + l^{\text{tr}} + l^{\text{ts}}$ inputs are sequentially provided. Among these, the initial $l^{\text{w}}$ inputs are disregarded to wash out the initial conditions, ensuring that the results remain independent of the chosen initial state. The subsequent $l^{\text{tr}}$ and $l^{\text{ts}}$ instances are used for training and testing, respectively, as detailed below. Since the input procedure involves a nonunitary alteration of the quantum state, we define a virtual time $\tau$, which is reset to zero at each input. Specifically, for $kt_{\text{in}} \leq t < (k+1)t_{\text{in}}$, $\tau$ is defined as $\tau \equiv t - kt_{\text{in}}$ [Fig. 2(c)]. The system subsequently undergoes time evolution under the Hamiltonian $\mathcal{H}$, simulated via the exact diagonalization method: $\rho(kt_{\text{in}} + \tau) = e^{-i\mathcal{H}\tau}\rho(kt_{\text{in}})e^{i\mathcal{H}\tau}$. For read-out, the expectation value of an operator $O$ is calculated as $\langle O(kt_{\text{in}} + \tau)\rangle = \text{Tr}[\rho(kt_{\text{in}} + \tau)O]$ [Fig. 2(b)]. We note that in the conventional QRC protocols, the virtual time is introduced to enhance expressivity by expanding the number of computational nodes (i.e., simultaneously processing observations at different time points) [13]. In contrast, the QRP framework leverages virtual time to provide temporal resolution to its computational capability. Specifically, performance is evaluated at different elapsed times after input ($\tau$), allowing the examination of how computational performance evolves over time. Hereafter, we denote $\langle O(kt_{\text{in}} + \tau)\rangle$ as $\langle O(\tau)\rangle$, unless the $k$ dependence is explicitly required.

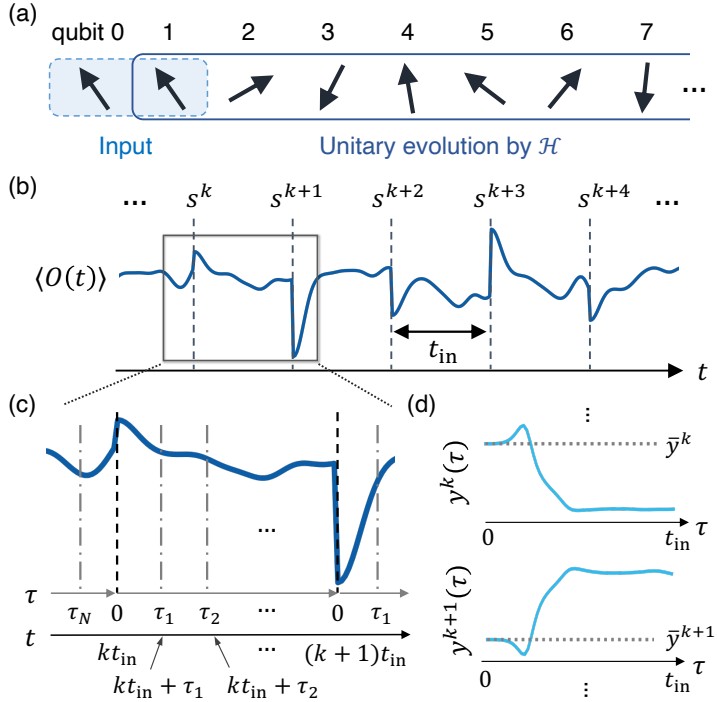

Figure 2: (a) Schematic of our quantum spin chain. Qubits 0 and 1 are simultaneously employed for input. The qubits $1, 2, \cdots$ evolve under the Hamiltonian in Eq. (5), while the qubit 0 is detached from the dynamics. (b) Quantum dynamics of the expectation value $\langle O(t) \rangle$ with input $\{s^k\}$ provided at time intervals $t_{\text{in}}$. (c) Concept of virtual time $\tau$. The expectation value $\langle O(\tau + kt_{\text{in}}) \rangle$ is used in the calculation of the output $y^k(\tau)$. (d) Dynamics of the output $y^k(\tau)$. The gray dotted line represents the target value $\bar{y}^k$. The performance at $\tau$ is evaluated based on the determination coefficient between $y(\tau)$ and $\bar{y}$.

As discussed in Sec. 2.1, the QRP captures information propagation through the capacity of a specific degree of freedom to estimate the input values. To elaborate, if the information of $s^k$ does not propagate to a degree of freedom $O(\tau)$, the original value $s^k$ cannot be estimated from $\langle O(\tau) \rangle$ at all; conversely, when propagated, $s^k$ can be accurately estimated from $\langle O(\tau) \rangle$. Following the QRC framework [13], this concept is formalized as the short-term memory (STM) task. The objective of this task is to produce the output $y^k(\tau)$ that accurately estimates the target $\bar{y}_d^k = s^{k-d}$, where $d$ denotes the delay steps. Using the read-out $\langle O(kt_{\text{in}} + \tau) \rangle$, the estimation output at the $k$-th step is calculated by a linear transformation as

$$y_d^k(\tau) = w_o(\tau) \langle O(kt_{\text{in}} + \tau) \rangle + w_c(\tau), \qquad (2)$$

where $w_o(\tau)$ and $w_c(\tau)$ are $k$-independent coefficients. For simplicity, we define an internal state vector $X^k(\tau) = (\langle O(kt_{\text{in}} + \tau) \rangle, 1)$ and a weight vector $\boldsymbol{w}(\tau) = (w_o(\tau), w_c(\tau))^\top$, yielding a concise representation of Eq. (2) as $y_d^k(\tau) = X^k(\tau) \boldsymbol{w}(\tau)$. The weight vector is optimized to produce the desired output using the training input dataset with $l^{\text{tr}}$ instances; subsequently, the estimation performance is evaluated on the unseen testing dataset with $l^{\text{ts}}$ instances.

Gathering internal state vectors in the training and testing phases, we construct an $(l^{\text{tr}} \times 2)$-dimensional matrix $X^{\text{tr}}(\tau) = \{X^k(\tau)\}_{k=l^{\text{w}}+1}^{l^{\text{w}}+l^{\text{tr}}}$ and an $(l^{\text{ts}} \times 2)$-dimensional matrix $X^{\text{ts}}(\tau) = \{X^k(\tau)\}_{k=l^{\text{w}}+l^{\text{tr}}+1}^{l^{\text{w}}+l^{\text{tr}}+l^{\text{ts}}}$, respectively. The corresponding target outputs for the STM task with delay $d$ are defined as an $l^{\text{tr}}$-dimensional vector $\bar{\boldsymbol{y}}_d^{\text{tr}} \equiv \{s^{k-d}\}_{k=l^{\text{w}}+1}^{l^{\text{w}}+l^{\text{tr}}}$ and an $l^{\text{ts}}$-dimensional vector $\bar{\boldsymbol{y}}_d^{\text{ts}} \equiv \{s^{k-d}\}_{k=l^{\text{w}}+l^{\text{tr}}+1}^{l^{\text{w}}+l^{\text{tr}}+l^{\text{ts}}}$. In the training phase, the weight vector is trained to minimize

the discrepancy between the target $\bar{\boldsymbol{y}}_d^{\text{tr}}$ and the output $\boldsymbol{y}_d^{\text{tr}}(\tau) = X^{\text{tr}}(\tau)\boldsymbol{w}(\tau)$ across all $k$ at each individual moment $\tau$ [Fig. 2(d)]. The optimal solution minimizing the least squared error is given by

$$\boldsymbol{w}(\tau) = X^{\text{tr}+}(\tau)\bar{\boldsymbol{y}}_d^{\text{tr}}, \tag{3}$$

where $X^{\text{tr}+}(\tau)$ denotes the Moore-Penrose pseudoinverse-matrix of $X^{\text{tr}}(\tau)$. In the testing phase, the estimation performance, i.e., the similarity between the target $\bar{\boldsymbol{y}}_d^{\text{ts}}$ and the testing output $\boldsymbol{y}_d^{\text{ts}}(\tau) = X^{\text{ts}}(\tau)\boldsymbol{w}(\tau)$, is evaluated using the determination coefficient

$$R_d^2(\tau) = \frac{\text{cov}^2(\boldsymbol{y}_d^{\text{ts}}(\tau), \bar{\boldsymbol{y}}_d^{\text{ts}})}{\sigma^2(\boldsymbol{y}_d^{\text{ts}}(\tau))\sigma^2(\bar{\boldsymbol{y}}_d^{\text{ts}})}, \tag{4}$$

where cov and $\sigma^2$ represent covariance and variance, respectively. $R_d^2(\tau)$ approaches one when the output $\boldsymbol{y}_d^{\text{ts}}(\tau)$ and the target $\bar{\boldsymbol{y}}_d^{\text{ts}}$ closely align; otherwise, it approaches zero. From this formalization of the QRP, the estimation performance functions as a quantitative metric to assess the extent to which the read-out operator $O(\tau)$ retains the information of the $d$-step previous input $s^{k-d}$. As shown below, our study primarily focuses on $R_{d=0}^2(\tau)$ to elucidate the mechanisms underlying the propagation of the most recently provided information. Notably, since only a linear transformation is applied to the raw expectation value, the resultant performance should accurately estimate the information stored in $O(\tau)$. Employing intricate transformations instead complicates the interpretation of the obtained performance, as the output reflects not only the physics associated with $\langle O(\tau)\rangle$ but also the intrinsic characteristics of the chosen transformation.

## 2.3 General remarks on QRP

Although Sec. 2.2 presented one particular QRP configuration optimized for probing information propagation, the QRP framework introduced in Sec. 2.1 is fundamentally more general. By adjusting its key components—task selection, read-out operator, input scheme, and performance metric—the QRP can be tailored to explore a broad spectrum of quantum many-body phenomena.

- **Task selection.** As the primary control parameter, the task determines which features of the quantum systems are highlighted by the QRP protocol. For instance, the STM task with $d = 0$, discussed in Sec. 2.2, is specifically suited for capturing information propagation dynamics. Conversely, tasks with finite delays ($d > 0$) can illuminate memory-related phenomena, including ergodicity [16]. Furthermore, incorporating nonlinear transformations in tasks enables the exploration of intrinsic nonlinear quantum effects.

- **Read-out operators.** The choice of read-out operator critically influences the QRP performance, as different quantum phenomena distinctively affect specific operators. Hence, the flexibility to freely choose read-out operators enables the QRP to probe quantum phenomena at operator-level resolution within the Hilbert space. While Sec. 3 primarily addresses local operators, employing non-local operators would be beneficial when exploring global characteristics of quantum systems.

- **Input schemes.** The input configuration defines the interface between quantum phenomena and information processing. Different input schemes enable distinct physical interpretations, as discussed in detail in Sec. 4. For instance, providing multiple inputs from distinct terminals allows the QRP to evaluate interactions among various excitations.

- **Performance metrics.** Performance metrics can be customized to align with specific research objectives. For example, the information processing capacity offers a systematic measure to probe nonlinear effects [50]. Alternatively, error-based metrics, such as mean squared error or mean absolute error, are suitable for quantifying system behavior under variations in internal or external parameters.

These flexible design choices lie at the heart of the QRP's power to explore diverse aspects of quantum many-body physics; for instance, the QRP has recently been employed to detect quantum phase transitions [51].

In practice, it is unrealistic to expect a single QRP configuration to be universally optimal for revealing all quantum phenomena. An unsuitable configuration may fail to capture the targeted physics, necessitating a systematic exploration of various setups to identify the optimal one. Such tuning procedure, however, is not a shortcoming specific to the QRP; rather, it represents a common and essential practice in any probing methodology throughout the physical sciences. Particularly in systems where prior knowledge is limited, systematically optimizing probing configurations is critical for extracting meaningful information. The inherent design flexibility of the QRP framework significantly facilitates this effort.

# 3 Applications: Information propagation in quantum systems

## 3.1 Ballistic and diffusive dynamics of information propagation

To demonstrate the effectiveness of the QRP, we investigate information propagation in a spin-1/2 Ising chain. The Hamiltonian is given by

$$\mathcal{H} = -J \sum_{i=1}^{N-1} \sigma_i^x \sigma_{i+1}^x + h_x \sum_{i=1}^{N} \sigma_i^x + h_z \sum_{i=1}^{N} \sigma_i^z, \tag{5}$$

with $h_z$ and $h_x$ representing the transverse and longitudinal magnetic fields, respectively. $\sigma_i^x$ and $\sigma_i^z$ are the $x$ and $z$ Pauli matrices at site $i$, and $J > 0$ is the strength of the nearest-neighbor Ising interaction, which we set to $J = 1$ as our energy scale. $N$ denotes the number of sites in the system, excluding the qubit 0, which is used as a reference ancilla when considering mutual information later and is therefore not involved in the time evolution [Fig. 2(a)]. The information $s^k$ is provided to the quantum system by setting the state of qubits 0 and 1 as $|\psi_{in}(s^k)\rangle = \sqrt{s^k}|00\rangle_{\{01\}} + \sqrt{1-s^k}|11\rangle_{\{01\}}$, following the scheme in Eq. (1). With the range $s^k \in [0,1]$, this input is, on average, symmetric with respect to the $x$ axis in spin space, effectively preserving the symmetry $\sigma_i^x \leftrightarrow -\sigma_i^x$ in the Hamiltonian at $h_x = 0$. This model is known to be mapped to a free fermion system via the Jordan-Wigner transformation in the case of $h_x = 0$, whereas it shows chaotic spectral statistics at $(h_x, h_z) = (-0.5, 1.05)$ [52]. We take $N = 7$, $t_{in} = 5$, and $(l^w, l^{tr}, l^{ts}) = (1000, 2000, 2000)$ in the following calculations.

Figure 3 represents the dynamics of the estimation performance $R_d^2(\tau)$ for $d = 0, 1, 2$ in two quantum systems: A free fermion system with $(h_x, h_z) = (0.0, 1.0)$ and a quantum chaotic system with $(h_x, h_z) = (-0.5, 1.05)$. For the calculation of the output $y^k(\tau)$, the expectation values $\langle \sigma_i^z(\tau) \rangle$ at each qubit $i$ are independently employed as the read-out operator. In other words, $R_d^2(\tau)$ in Fig. 3 quantifies the information propagated to the $z$ component of individual spins at each moment. Immediately after the input, i.e., when $\tau \ll 1$, the information $s^k$ remains predominantly localized within the qubit 1 where the input is provided. This is evidenced by the almost unity $R_{d=0}^2(\tau \simeq 0)$ for the qubit 1, while being vanishingly small for the remaining qubits. Subsequently, the information propagates through qubits 2, 3, ..., leading to a gradual emergence of nonzero values for $R_{d=0}^2(\tau)$ from the qubits close to the qubit 1. At

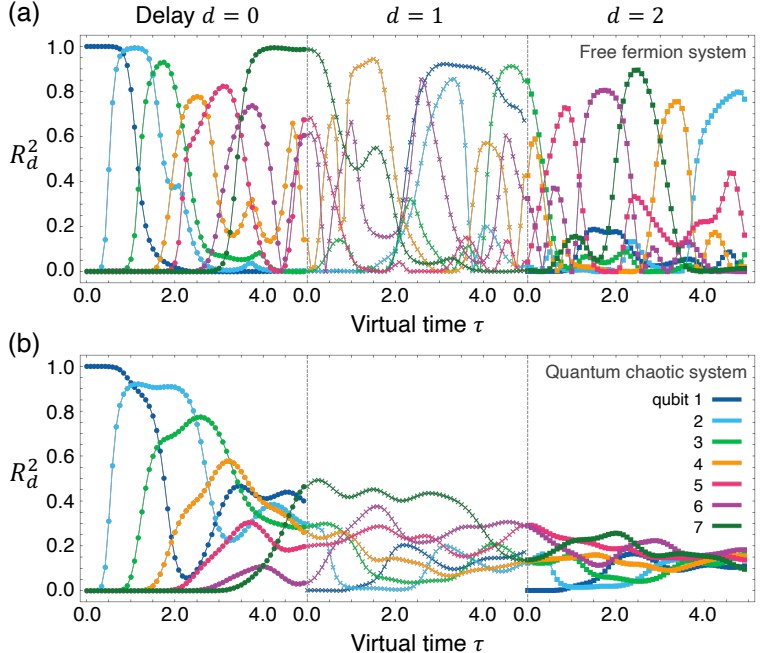

Figure 3: (a) Estimation performance $R_d^2(\tau)$ in the STM task with delays $d = 0, 1, 2$ in the free fermion system with $h_x = 0.0$ and $h_z = 1.0$. $\langle \sigma_i^z(\tau) \rangle$ is utilized in the calculation of the output $y_d^k(\tau)$. The colors represent each qubit, and the marker styles indicate different delays. (b) The same plot as in (a) for the quantum chaotic system with $h_x = -0.5$ and $h_z = 1.05$.

$\tau = t_{\text{in}}$ ($t = (k+1)t_{\text{in}}$), new information $s^{k+1}$ is provided to the input qubits. The information of $s^k$ remaining within the quantum reservoir system is then evaluated via the STM task with $d = 1$. Upon this input operation, the quantum state of the input qubits undergoes a substantial alteration, while the states of the other qubits remain largely unchanged. Indeed, $R_{d-1}^2(\tau \to t_{\text{in}})$ and $R_d^2(\tau = 0)$ exhibit continuous connectivity, except for the qubit 1, which is designated for input (Fig. 2). $R_d^2(\tau)$ thus effectively corresponds to $R_{d=0}^2(\tau + t_{\text{in}}d)$ under the condition where inputs are successively provided.

Remarkably, the nature of information propagation is closely linked to the underlying dynamics of the quantum system. In the case of the free fermion system, information propagates ballistically, as illustrated in Fig. 3(a). The dynamics of $R_{d=0}^2(\tau)$ for each qubit exhibits a unimodal behavior, with peaks sequentially moving to neighboring sites. This ballistic behavior signifies quasiparticle-mediated information propagation. On the quenching process that introduces new input information, a quasiparticle carrying the provided information is excited at the input qubits. As the quasiparticle traverses along the chain, the peak of $R_{d=0}^2(\tau)$, representing the most recently provided information, moves to the qubit where the quasiparticle exists. Such localized behavior gives rise to the unimodal shape observed in Fig. 3(a). Proceeding to the next input, a new quasiparticle is created and interferes with the existing ones. As a result, $R_{d=1}^2(\tau)$ and $R_{d=2}^2(\tau)$ exhibit relatively complicated dynamics, albeit their ballistic nature is similar to that of $d = 0$.

In contrast, Fig. 3(b) supports diffusive information propagation in the quantum chaotic system. The timeline for $R_{d=0}^2(\tau)$ commencing its ascension at each qubit is similar to that in Fig. 3(a); however, the process of information accumulation toward the maximum of $R_{d=0}^2(\tau)$ proves to be significantly prolonged. In addition, after reaching its maximum, $R_{d=0}^2(\tau)$ exhibits a gradual decay over a timescale of $t_{\text{in}}$, which contrasts with an abrupt post-peak decline

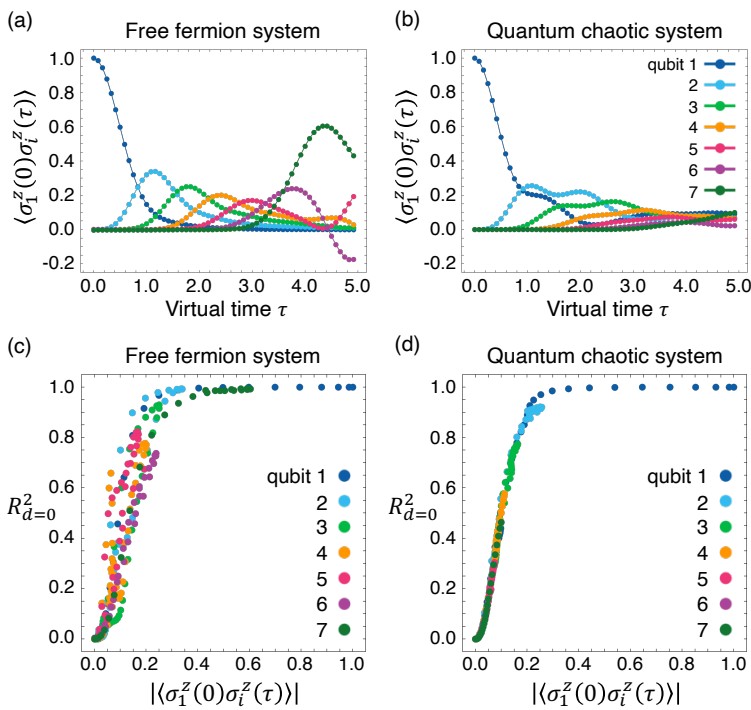

Figure 4: (a)-(b) The dynamical spin correlation between qubit 1 and each qubit $i$, averaged over the testing inputs. (c)-(d) The relationship between the dynamical spin correlation $|\langle \sigma_1^z(0)\sigma_i^z(\tau)\rangle|$ and the estimation performance $R_{d=0}^2(\tau)$ obtained from $\langle \sigma_i^z \rangle$. Each qubit is represented by distinct colors. (a), (c) Free fermion system and (b), (d) quantum chaotic system.

on a timescale of $\Delta\tau \sim 2$ observed in the free fermion system. Both $R_{d=1}^2(\tau)$ and $R_{d=2}^2(\tau)$ similarly demonstrate smooth and gradual dynamics without pronounced peaks. Notably, $R_{d=2}^2(\tau \to t_{\text{in}})$ converges toward a uniform value across all qubits, implying a homogeneous spread of information throughout all the qubits owing to the information delocalization. In Appendix A, we present the dynamics of $R_d^2(\tau)$ with varying system sizes. Therein, the QRP captures qualitatively the same behavior as Fig. 3, emphasizing the generality of ballistic or diffusive information propagation in each system irrespective of system size.

For further elucidation of the mechanisms of information propagation, we examine the dynamical spin correlation between the individual qubits and the input qubit 1, $\langle \sigma_1^z(0)\sigma_i^z(\tau)\rangle$, in Figs. 4(a) and 4(b). In contrast to the statistically defined $R_d^2$, physical observables, including the correlations, depend on the individual input value $s^k$. Henceforth, we utilize the mean value over the testing input instances when considering physical observables. In the free fermion system, the dynamical spin correlation for $i \geq 2$ initiates an ascension, achieves its maximum, and thereafter undergoes an attenuation; this entire process occurs sequentially, ordered by the distance from qubit 1 [Fig. 4(a)]. Conversely, in the quantum chaotic system in Fig. 4(b), the dynamical spin correlation accumulates progressively over time and maintains a value of approximately 0.1 for long periods.

Figures 4(c) and 4(d) illustrate the relationship between the dynamical spin correlation $|\langle \sigma_1^z(0)\sigma_i^z(\tau)\rangle|$ and the estimation performance $R_{d=0}^2(\tau)$ when individual $\langle \sigma_i^z(\tau)\rangle$ is employed as the read-out. As suggested by previous studies on classical magnetic physical reservoirs [12, 53, 54], the quantum reservoir system achieves higher $R_{d=0}^2(\tau)$ when the spin variable harnessed as the read-out is strongly correlated with the input spin $\sigma_1^z(0)$. Remarkably, despite the intricate dynamics displayed by both quantities in the quantum chaotic sys-

tem [Figs. 3(b) and 4(b)], the data collapse onto a single curve in Fig. 4(d), indicating an almost one-to-one correspondence between $|\langle\sigma_1^z(0)\sigma_i^z(\tau)\rangle|$ and $R_{d=0}^2(\tau)$ irrespective of the qubit position $i$ and virtual time $\tau$. This is accentuated by comparison with the more dispersed plot for the free fermion system in Fig. 4(c). We quantify the deviation from a perfect one-to-one correspondence between these two quantities using the data deviation criterion $\Delta$. For a given integer $0 \le m \le M-1$, we define $\Lambda_m$ as the set of $\{(i,\tau)\}$ that satisfy $m/M \le |\langle\sigma^z(0)\sigma^z(\tau)\rangle| < (m+1)/M$, where $M$ represents the number of windows (we set $M = 4000$). The average of $R_{d=0}^2$ over $\Lambda_m$ is denoted as $\overline{\left(R_{d=0}^2\right)}_m$. Under the assumption of the one-to-one correspondence, $R_{d=0}^2$ calculated for each $(i,\tau) \in \Lambda_m$ should be close to this average. The data deviation $\Delta$ is then defined as the summation of the squared deviations from the average, given by $\Delta \equiv \sum_{m=0}^{M-1}\sum_{(i,\tau)\in\Lambda_m}\left[\left(R_{d=0}^2\right)_{(i,\tau)} - \overline{\left(R_{d=0}^2\right)}_m\right]^2$. In the quantum chaotic system, the deviation criterion $\Delta$ is evaluated to be $\Delta \simeq 0.0299$, which is approximately one order of magnitude smaller than the value of $\Delta \simeq 0.2866$ for the free fermion system. This quantitative assessment substantiates the one-to-one correspondence between the spin correlation and the estimation performance in the former system. In contrast to the ballistic propagation mediated by quasiparticles in the free fermion system, this observation suggests that the spin correlations play a pivotal role in the diffusive information propagation in the quantum chaotic system.

## 3.2 Information propagation channels in the Hilbert space

We here emphasize that the QRP possesses the capability to assess information propagation to any arbitrary operator $O(\tau)$. The estimation performance $R_d^2(\tau)$, derived from the output $y(\tau)$ obtained through the linear transformation of $\langle O(\tau)\rangle$, serves as a quantitative measure of the information stored in that degree of freedom. By systematically scanning different read-out operators, the QRP can explore the spread of information across multiple degrees of freedom in the Hilbert space at any given moment, thereby identifying specific channels for information propagation.

Figures 5(a) and 5(c) represent $R_{d=0}^2(\tau)$ employing observables of the single spin $\left(\langle\sigma_2^z(\tau)\rangle\right.$ and $\left.\langle\sigma_3^z(\tau)\rangle\right)$ and spin correlation $\left(\langle\sigma_2^x(\tau)\sigma_3^x(\tau)\rangle\right.$ and $\left.\langle\sigma_2^z(\tau)\sigma_3^z(\tau)\rangle\right)$. In the free fermion system [Fig. 5(a)], $R_{d=0}^2(\tau)$ initially increases at the qubit 2, and before it becomes nonzero for the qubit 3, the information propagates to the $x$ component of the correlation between the qubits 2 and 3: $\langle\sigma_2^x(\tau)\sigma_3^x(\tau)\rangle$. However, $R_{d=0}^2(\tau)$ for the $z$ component of the correlation $\langle\sigma_2^z(\tau)\sigma_3^z(\tau)\rangle$ remains nearly zero at all time. Detailed results utilizing other operators are presented in Appendix B, yet it is pertinent to note that $R_{d=0}^2(\tau)$ manifests a nonzero value only when employing the $z$ spin on a single site $\langle\sigma_i^z(\tau)\rangle$ or the $x$ component of the nearest-neighbor spin correlation $\langle\sigma_i^x(\tau)\sigma_{i+1}^x(\tau)\rangle$. These observations unequivocally indicate that the information propagates through the channel of spin $x$ interactions between nearest qubits. This is consistent with the picture of quasiparticle-mediated information propagation, as the interaction $\sigma_i^x\sigma_{i+1}^x$ constitutes the foundation of the quasiparticle description in the free fermion system.

In contrast, in the quantum chaotic system, both the $x$ and $z$ components of correlations retain information, as indicated by nonzero $R_{d=0}^2(\tau)$ in Fig. 5(c). This behavior also extends to other operators, including correlations among distant qubits (Appendix B). In particular, $R_{d=0}^2(\tau)$ using $\langle\sigma_2^x(\tau)\sigma_3^x(\tau)\rangle$ and $\langle\sigma_2^z(\tau)\sigma_3^z(\tau)\rangle$ exhibit similar behavior at early time, and as time evolves, they diverge and display different behaviors. Each type of spin correlation thus serves as an individual channel for information propagation between adjacent qubits. This marks a significant distinction from the free fermion system, where the information propagation channels are limited to a few correlations. Notably, although our investigations focus on the early time regimes, the nonzero $R_d^2$ observed in various degrees of freedom (Appendix B)

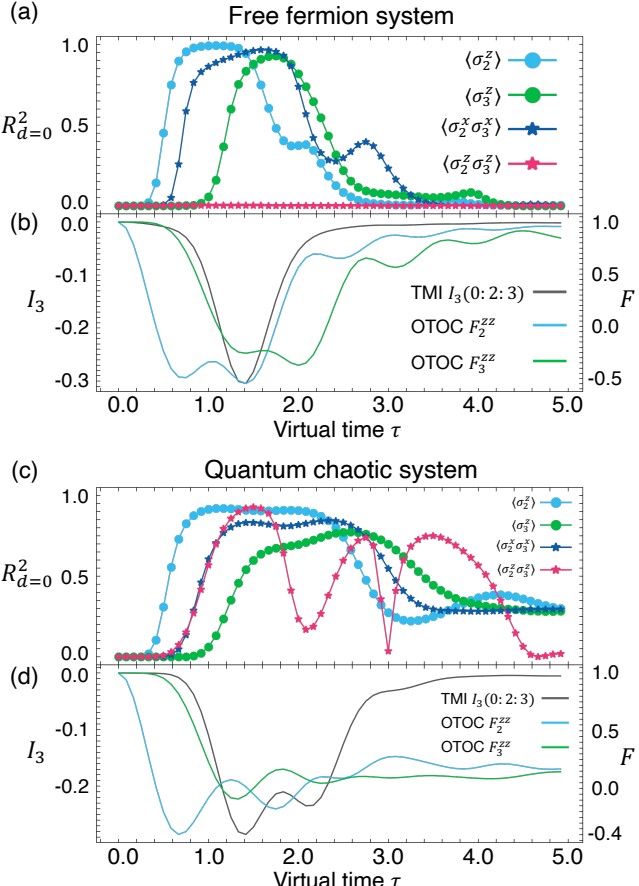

Figure 5: (a), (c) Dynamics of the estimation performance in the QRP framework. The skyblue (green) line represents $R^2_{d=0}(\tau)$ calculated using the single spin $\langle\sigma^z_2(\tau)\rangle$ $\left(\langle\sigma^z_3(\tau)\rangle\right)$, whereas the blue (pink) line illustrates $R^2_{d=0}(\tau)$ when the spin correlation $\langle\sigma^x_2(\tau)\sigma^x_3(\tau)\rangle$ $\left(\langle\sigma^z_2(\tau)\sigma^z_3(\tau)\rangle\right)$ is utilized. (b), (d) Dynamics of the OTOC and TMI averaged over the testing inputs. The skyblue and green lines plot the OTOC for qubit 2 ($F^{zz}_2(\tau)$) and for qubit 3 ($F^{zz}_3(\tau)$), respectively. The black line displays the TMI among qubits 0, 2, and 3. (a)-(b) Free fermion system, and (c)-(d) quantum chaotic system.

could be considered an early signature of quantum information scrambling in the long-time limit, where information delocalizes over diverse degrees of freedom.

## 3.3 Comparisons with OTOC and TMI

In the previous sections, we have explored information propagation via the estimation performance using the QRP. To validate its reliability, we here compare the QRP with conventional methodologies for evaluating information propagation, namely, the out-of-time-order correlator (OTOC) and the tripartite mutual information (TMI). The OTOC essentially probes the degree of noncommutativity between two initially commuting operators at different temporal points [55–61]. The long-time behavior of the OTOC, particularly its asymptotic value, is a key indicator of the presence or absence of scrambling [62–65]. We specifically calculate the OTOC between the qubits $i$ and 1 as $F^{zz}_i \equiv \langle\sigma^z_i(\tau)\sigma^z_1(0)\sigma^z_i(\tau)\sigma^z_1(0)\rangle$. On the other hand, the TMI quantifies the extent to which information about one subsystem can be extracted from the nonlocal correlations present between two other subsys-

tems [32, 66–68]. Defined in an operator-independent manner, it becomes negative when the targeted information delocalizes across the subsystems. Here, we utilize the detached input qubit 0 as the reference system for $s^k$, and evaluate the spread of information of $s^k$ over the nonlocal correlations between the qubits 2 and 3. The corresponding TMI is defined as $I_3(0\colon 2\colon 3) \equiv S_{\{0\}} + S_{\{2\}} + S_{\{3\}} - S_{\{0\}\cup\{2\}} - S_{\{0\}\cup\{3\}} - S_{\{2\}\cup\{3\}} + S_{\{0\}\cup\{2\}\cup\{3\}}$, where $S_X$ is the von Neumann entropy. Both the OTOC and TMI are averaged over the testing input instances.

Figures 5(b) and 5(d) illustrate the OTOC and TMI in the free fermion system and the quantum chaotic system. In the initial stage, the OTOC $F_i^{zz}(\tau)$ for the qubits 2 and 3 begin to decrease, slightly before the rise of $R_{d=0}^2(\tau)$ utilizing $\langle\sigma_2^z(\tau)\rangle$ and $\langle\sigma_3^z(\tau)\rangle$, respectively. During the intermediate stage, the TMI $I_3(0\colon 2\colon 3)$ turns negative at the same time as $R_{d=0}^2(\tau)$, calculated from the spin correlations, become nonzero. Both indicate initial spread of input information over the qubits 2 and 3 in that time regime, which completely aligns with the behavior of $R_d^2(\tau)$ in the QRP [Figs. 5(a) and 5(c)]. As $\tau$ approaches $t_{\mathrm{in}}$, the OTOC $F_i^{zz}(\tau)$ converges to 1 in the free fermion system and to 0 in the quantum chaotic system. This long-time asymptotic value signifies the absence or presence of scrambling in each system [32, 58], which is also consistent with whether or not the nonzero $R_d^2(\tau)$ spreads for various operators in the QRP. These parallel observations validate the reliability of the QRP in capturing information propagation in quantum systems.

Fundamentally, the dynamics in the free fermion system and the quantum chaotic system are qualitatively distinct. Beyond differentiating the ballistic and diffusive propagation dynamics of $R_{d=0}^2(\tau)$ (Fig. 3), the QRP elucidates these disparities from the perspective of the information propagation channels: the pronounced distinction in $R_{d=0}^2(\tau)$ derived from $\langle\sigma_2^z(\tau)\sigma_3^z(\tau)\rangle$ offers compelling evidence of the differences in the propagation channels [Figs. 5(a) and 5(c)]. However, such differences in propagation dynamics cannot be deduced from the behaviors of OTOC or TMI, as illustrated in Figs. 5(b) and 5(d). The OTOC in these systems differ in their asymptotic values, while their early and intermediate dynamics remain notably similar, offering little insight into the information propagation channels (nor can the OTOC for other operator pairs in Appendix C). The TMI displays qualitatively similar dynamics between these systems throughout the entire temporal regime. Its operator-independent definition obscures the influence of specific degrees of freedom that differentiate these quantum systems. Consequently, the fundamental strength of the QRP lies in its resolution to analyze information propagation for arbitrary degrees of freedom at any specific point in time. This in-depth analysis effectively uncovers the intrinsic dynamical characteristics of quantum systems, including the underlying information propagation channels. Moreover, it is worth highlighting the greater experimental feasibility of the QRP, as it solely requires expectation values of pertinent operators, such as spins and spin correlations. This stands in stark contrast to OTOC, which requires inverse time evolution, and to TMI, which necessitates highly precise quantum state tomography [58, 69–72].

We further investigate the perturbed system with $(h_x, h_z) = (-0.02, 1.002)$ to lucidly demonstrate the sensitivity of the QRP. These parameters closely approximate those of the free fermion system; however, the system is no longer integrable, and the symmetry $\sigma_i^x \leftrightarrow -\sigma_i^x$ is broken. Figures 6(a) and 6(b) show $R_{d=0}^2(\tau)$ when each of the following is employed as the read-out operator in the free fermion system and the perturbed system, respectively: $\langle\sigma_2^x(\tau)\sigma_3^x(\tau)\rangle$, $\langle\sigma_2^z(\tau)\sigma_3^z(\tau)\rangle$, $\langle\sigma_2^z(\tau)\sigma_3^x(\tau)\rangle$, and $\langle\sigma_2^x(\tau)\sigma_3^z(\tau)\rangle$. Due to the similarity of the models, $R_{d=0}^2(\tau)$ employing $\langle\sigma_2^x(\tau)\sigma_3^x(\tau)\rangle$ and $\langle\sigma_2^z(\tau)\sigma_3^z(\tau)\rangle$ are semiquantitatively indistinguishable between these two systems. Nevertheless, the breakdown of the symmetry and quasiparticle picture due to the perturbation gives rise to different types of information propagation channels beyond quasiparticle mediation, as indicated by $R_{d=0}^2(\tau)$ utilizing $\langle\sigma_2^z(\tau)\sigma_3^x(\tau)\rangle$ and $\langle\sigma_2^x(\tau)\sigma_3^z(\tau)\rangle$, which only display nonzero values in the perturbed system [Fig. 6(b)].

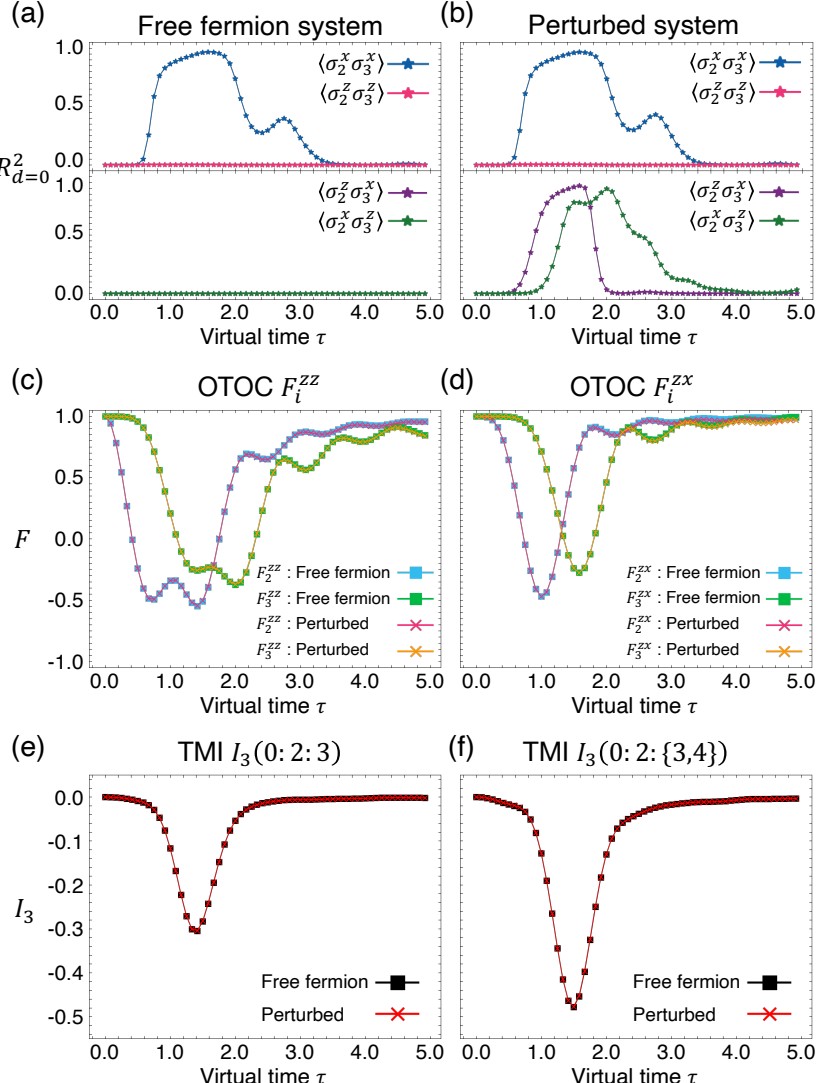

Figure 6: (a) The estimation performance $R^2_{d=0}(\tau)$ in the free fermion system with $(h_x, h_z) = (0.0, 1.0)$, employing the read-out operator $\langle\sigma^x_2(\tau)\sigma^x_3(\tau)\rangle$ (blue), $\langle\sigma^z_2(\tau)\sigma^z_3(\tau)\rangle$ (pink), $\langle\sigma^z_2(\tau)\sigma^x_3(\tau)\rangle$ (purple), and $\langle\sigma^x_2(\tau)\sigma^z_3(\tau)\rangle$ (green). (b) The same plot as (a) in the perturbed system with $(h_x, h_z) = (-0.02, 1.002)$. (c)-(d) The OTOC for qubits 2 and 3: (c) $F^{zz}_i$ and (d) $F^{zx}_i$. The skyblue and green lines represent the OTOC in the free fermion system, while the pink and orange lines correspond to the perturbed system. (e)-(f) Dynamics of the TMI: (e) $I_3(0:2:3)$ and (f) $I_3(0:2:\{3,4\})$. The black and red lines plot the TMI in the free fermion system and the perturbed system, respectively.

In Figs. 6(c) and 6(d), we illustrate the dynamics of OTOC $F^{zz}_i(\tau)$ and similarly defined $F^{zx}_i(\tau)$. Remarkably, despite the qualitative differences between the free fermion system and the perturbed system, these OTOC manifest almost identical values in both systems, as evidenced by the overlapping pairs of curves (similar agreements are also observed for $F^{xx}_i(\tau)$ and $F^{xz}_i(\tau)$). We also present the TMI $I_3(0:2:3)$ and $I_3(0:2:\{3,4\})$ in Figs. 6(e) and 6(f) respectively, with the latter defined analogously to the former. As in the case of the OTOC, the overlapping curves therein underscore the incapacity of the TMI to distinguish between the two systems. These observations highlight the marked disparity in sensitivity between

the QRP and the OTOC or TMI, as the latter two exhibit less responsiveness to perturbations, even those involving changes in symmetry or integrability. The pronounced sensitivity of the QRP facilitates a detailed investigation of quantum many-body physics from an informational perspective, which might remain obscured in conventional analyses using the OTOC and TMI.

# 4 Discussion and conclusion

In this paper, we have proposed the QRP by inversely extending the QRC for the exploration of quantum many-body physics from the perspectives of computation and information. By establishing a correspondence between the physical properties and the computational performance, the QRP can shed light on the physics in any degree of freedom at arbitrary time. Among many potential applications of the QRP, we have concentrated on the study of information propagation within the Hilbert space. Here, sequential input information is provided to the quantum reservoir system via the local quantum quench, subsequently estimated using various read-out operators. The estimation performance is utilized as the metric for information propagation. In the quantum Ising chain with transverse and longitudinal magnetic fields, we have demonstrated that the QRP captures both ballistic information propagation mediated by quasiparticles in the free fermion system and diffusive information propagation facilitated by spin correlations in the quantum chaotic system, with the latter exhibiting early signatures of information scrambling across various degrees of freedom. Furthermore, we have shown that the QRP can systematically identify system-specific information propagation channels through a comprehensive scan of read-out operators, which is an advantage over conventional measures, in addition to its pronounced sensitivity to perturbations.

Through the examination of information propagation, the QRP offers a powerful framework for uncovering the relationship between specific operations and their resultant quantum dynamics. Conventional approaches to understanding the impact of operations, such as sudden quenches, application of electromagnetic fields, or coupling with external systems, typically involve the direct observation of physical observables under the influence of these operations. However, the resulting dynamics is often affected by other multiple intrinsic and extrinsic factors, the complexity of which precludes straightforward inference of the underlying causal relationships. In contrast, the QRP conceptualizes quantum dynamics as a process that conveys quantum information throughout the system. Particularly in this study, where the input is initially provided via the quantum quench operation, the propagation of the input information can be considered equivalent to the propagation of quenching effects, with quasiparticles or quantum correlations mediating this process. Similarly, when the input originates from, for example, magnetic fields or electric currents, information propagation can be interpreted as the spread of the effects caused by these applied fields or currents. In this way, the QRP characterizes an operation as the source of input information, and the spread of its effects is captured through the propagation of that information. This enables the QRP to selectively extract target effects in isolation from other influences of diverse origins. Such a methodology would prove invaluable in a wide range of contexts for analyzing phenomena of interest without being obscured by the complex interplay of various factors.

Finally, we emphasize the broad applicability of the QRP, which extends beyond the analysis of information propagation. As discussed in Sec. 2.3, the QRP can investigate diverse properties of quantum systems through customized configurations, including input schemes, target outputs, evaluation metrics, and read-out observables. Moreover, the QRP is applicable to arbitrary systems, and its potential applications to high-dimensional, dissipative, or topological systems promise to yield further insights into largely unexplored quantum many-body phenomena, including mesoscopic, non-Hermitian, and topological quantum physics. It is worth

noting that the QRP can be implemented using the same experimental configuration as the QRC, which has already been successfully realized in several systems [27–29]. Potential platforms include optical lattices [73], photonic simulators [74], and trapped ions [75], spanning a wide range of quantum architectures. Considering its design flexibility, broad applicability, and experimental feasibility, we believe the QRP will establish itself as a potent tool for further propelling the exploration of quantum many-body physics.

## Acknowledgments

We thank Yasuyuki Kato for fruitful discussions.

**Funding information**    This research was supported by a Grant-in-Aid for Scientific Research on Innovative Areas "Quantum Liquid Crystals" (KAKENHI Grant No. JP19H05825) from JSPS of Japan and JST CREST (No. JP-MJCR18T2). K.K. was supported by the Program for Leading Graduate Schools (MERIT-WINGS), JSPS KAKENHI Grant Number JP24KJ0872, and JST BOOST Grant Number JPMJBS2418.

## A    Information propagation dynamics with varying system sizes

We examine the system size dependence of the information propagation dynamics. Figure 7 extends Fig. 3 by showcasing the estimation performance $R_d^2(\tau)$ calculated using individual spin operators $\langle \sigma_i^z(\tau) \rangle$ for system sizes ranging from $N = 6$ to $10$.

In the free fermion system, we observe characteristic sequential peaks in $R_{d=0}^2$, exhibiting ballistic propagation from qubit 1 toward qubits 2, 3, and so forth. Conversely, the quantum chaotic system demonstrates diffusive propagation of $R_{d=0}^2$ throughout the system. These qualitative behaviors remain consistent across different system sizes, suggesting the applicability of the QRP for capturing the characteristics of information propagation dynamics independent of system size.

## B    Read-out with various degrees of freedom

We investigate the information stored in various types of operators using the QRP protocol. Specifically focusing on the information contained in qubits 2, 3, and 4, we calculate the estimation performance by employing the spin $\sigma_i^{(x,z)}(\tau)$ and spin correlations $\sigma_i^{(x,z)}(\tau)\sigma_j^{(x,z)}(\tau)$ over these qubits. Figure 8 illustrates $R_{d=0}^2(\tau)$ for three different systems: the free fermion system with $(h_x, h_z) = (0.0, 1.0)$, the quantum chaotic system with $(h_x, h_z) = (-0.5, 1.05)$, and the perturbed system with $(h_x, h_z) = (-0.02, 1.002)$.

In the free fermion system, $R_{d=0}^2(\tau)$ exhibits a nonzero value when either $\langle \sigma_i^z(\tau) \rangle$ [Fig. 8(d)] or $\langle \sigma_i^x(\tau)\sigma_{i+1}^x(\tau) \rangle$ [Figs. 8(g) and 8(m)] is utilized as the read-out operator. Otherwise, $R_{d=0}^2(\tau)$ becomes nearly zero, indicating that information is not stored in operators such as $\langle \sigma_i^x(\tau) \rangle$ [Fig. 8(a)], $\langle \sigma_i^z(\tau)\sigma_j^z(\tau) \rangle$ [Figs. 8(g), 8(m), and 8(s)], $\langle \sigma_i^x(\tau)\sigma_j^z(\tau) \rangle$, $\langle \sigma_i^z(\tau)\sigma_j^x(\tau) \rangle$ [Figs. 8(j), 8(p), and 8(v)], and $\langle \sigma_i^x(\tau)\sigma_{j \neq i+1}^x(\tau) \rangle$ [Fig. 8(s)]. We note that due to the inherent symmetry $\sigma_i^x \leftrightarrow -\sigma_i^x$ in the free fermion system, the expectation values of odd operators with respect to $\sigma_i^x$ vanish, resulting in $R_{d=0}^2(\tau) \simeq 0$ when utilizing such operators.

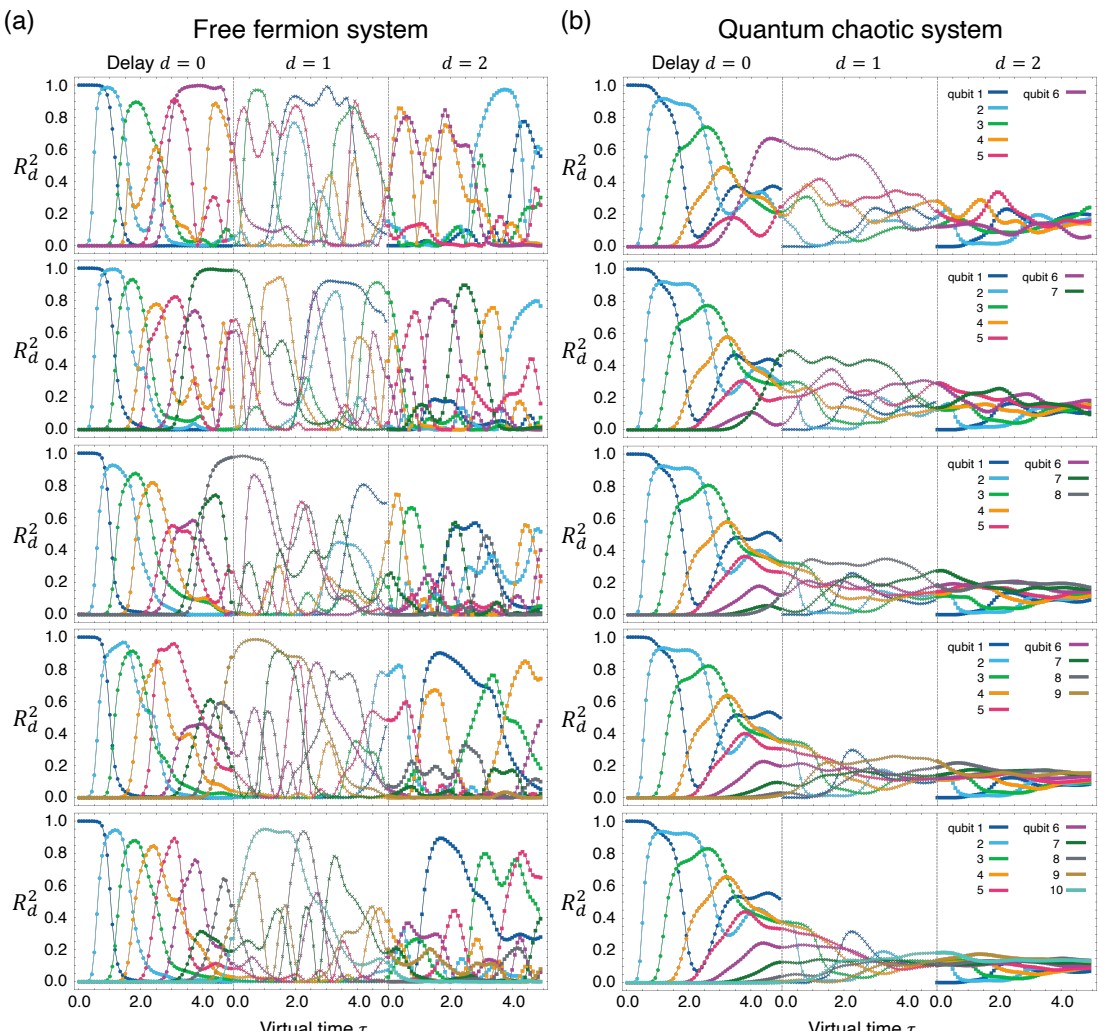

Figure 7: (a) System size dependence of the estimation performance $R_d^2(\tau)$ in the STM task with delays $d = 0, 1, 2$ in the free fermion system with $h_x = 0.0$ and $h_z = 1.0$. The expectation value $\langle\sigma_i^z(\tau)\rangle$ is utilized in the calculation of the output $y(\tau)$. From the top to bottom, the system size ranges from $N = 6$ to 10. (b) The same plot as in (a) for the quantum chaotic system with $h_x = -0.5$ and $h_z = 1.05$. The colors represent each qubit, and the marker styles indicate different delays.

In the quantum chaotic case, $R_{d=0}^2(\tau)$ manifests nonzero values for all the read-out operators shown in Fig. 8, including correlations between qubits 2 and 4, despite the distance between them [Figs. 8(u) and 8(x)]. This represents an early signature of quantum information scrambling, where information diffuses across a multitude of degrees of freedom.

In the perturbed system, $R_{d=0}^2(\tau)$ employing $\langle\sigma_i^z(\tau)\rangle$ and $\langle\sigma_i^x(\tau)\sigma_{i+1}^x(\tau)\rangle$ are semiquantitatively similar to those in the free fermion system, as evidenced by the almost identical pairs of figures: Figs. 8(d) and 8(e), Figs. 8(g) and 8(h), and Figs. 8(m) and 8(n). However, due to the breakdown of the symmetry and the quasiparticle picture, information propagates across a broader range of degrees of freedom. Indeed, the $x$ component of each spin $\langle\sigma_i^x(\tau)\rangle$ manifests nonzero $R_{d=0}^2(\tau)$ [Fig. 8(b)], and the spin correlations $\langle\sigma_i^x(\tau)\sigma_j^z(\tau)\rangle$ and $\langle\sigma_i^z(\tau)\sigma_j^x(\tau)\rangle$ [Figs. 8(k), 8(q), and 8(w)] also exhibit nonzero $R_{d=0}^2(\tau)$. The latter suggests that these correlations serve as additional information propagation channels between qubits $i$ and $j$, alongside quasiparticle mediation.

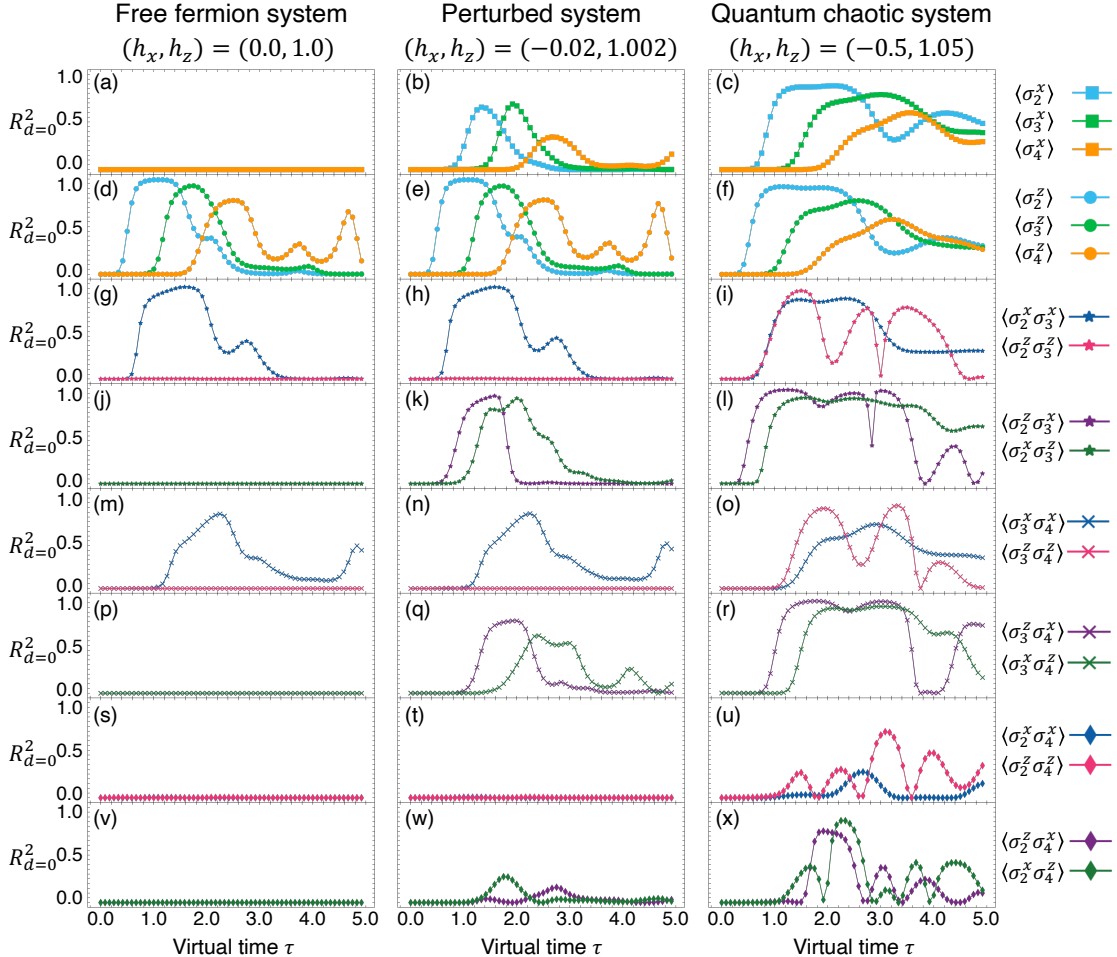

Figure 8: Dynamics of the reservoir performance $R^2_{d=0}$ for the STM task employing various operators of qubits 2, 3, 4 for the read-out. The legends on the rightmost side show the correspondence between markers and operators. (a, d, g, j, m, p, s, v) Free fermion system, (b, e, h, k, n, q, t, w) perturbed system, and (c, f, i, l, o, r, u, x) quantum chaotic system.

## C   OTOC for various operator pairs

In Fig. 5(a), we demonstrate that $R^2_{d=0}$ using $\langle \sigma^x_2(\tau)\sigma^x_3(\tau)\rangle$ and $\langle \sigma^z_2(\tau)\sigma^z_3(\tau)\rangle$ exhibit markedly distinct behaviors in the free fermion system, which indicates that information propagates primarily through $\sigma^x_2(\tau)\sigma^x_3(\tau)$, rather than through $\sigma^z_2(\tau)\sigma^z_3(\tau)$. Conversely, in the quantum chaotic system [Fig. 5(c)], $R^2_{d=0}$ for both read-out operators becomes finite, suggesting that each type of spin correlation serves as an independent information propagation channel. However, the OTOC $F^{zz}_2 = \langle \sigma^z_2(\tau)\sigma^z_1(\tau)\sigma^z_2(\tau)\sigma^z_1(0)\rangle$ and $F^{zz}_3 = \langle \sigma^z_3(\tau)\sigma^z_1(\tau)\sigma^z_3(\tau)\sigma^z_1(0)\rangle$ exhibit similar dynamics between these systems, except for the asymptotic value, as shown in Figs. 5(b) and 5(d). This suggests the inability of the OTOC to identify information propagation channels in the Hilbert space.

To further illustrate this limitation, we examine the OTOC for various operator pairs, specifically focusing on the qubits 2 and 3. In Figs. 9(a), 9(b), and 9(c), corresponding respectively to the free fermion system, the perturbed system, and the quantum chaotic system, we illustrate the OTOC between the qubit 1 and the correlations of the qubits 2 and 3: $F^{xx,z}_{23,1} = \langle (\sigma^x_2\sigma^x_3)(\tau)\sigma^z_1(0)(\sigma^x_2\sigma^x_3)(\tau)\sigma^z_1(0)\rangle$ and $F^{zz,z}_{23,1} = \langle (\sigma^z_2\sigma^z_3)(\tau)\sigma^z_1(0)(\sigma^z_2\sigma^z_3)(\tau)\sigma^z_1(0)\rangle$.

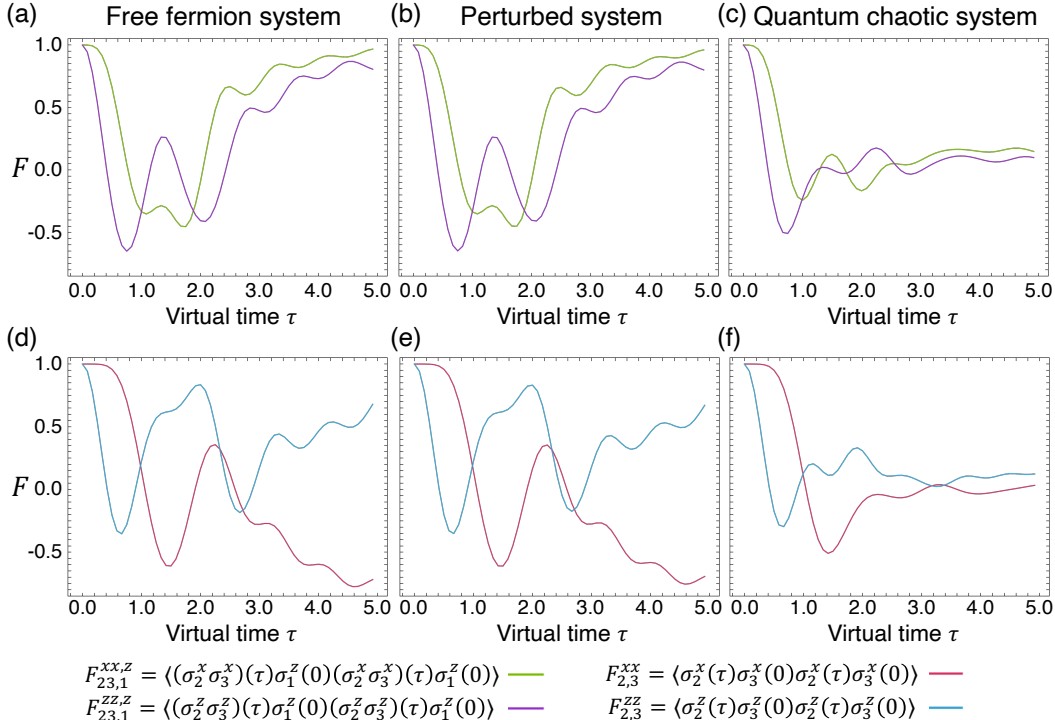

Figure 9: (a)-(c) Dynamics of the OTOC averaged over the testing inputs: $F_{23,1}^{xx,z}$ (green) and $F_{23,1}^{zz,z}$ (purple). (d)-(f) The same plot for $F_{2,3}^{xx}$ (pink) and $F_{2,3}^{zz}$ (skyblue). (a), (d) Free fermion system, (b), (e) perturbed system, and (c), (f) quantum chaotic system.

Similar to the observations in Fig. 6, the OTOC for the first two systems exhibit semiquantitative similarity, while those for the quantum chaotic system deviate in the asymptotic values, corresponding to the occurrence of scrambling. However, other qualitative differences, particularly regarding information propagation channels, are not inferred from them. Even when comparing the two OTOC within the free fermion system [Fig. 9(a)], $F_{23,1}^{xx,z}$ and $F_{23,1}^{zz,z}$ display similar behavior, failing to reveal any qualitative differences between $\sigma_2^x(\tau)\sigma_3^x(\tau)$ and $\sigma_2^z(\tau)\sigma_3^z(\tau)$, contrary to results obtained with the QRP that suggest distinct roles for these operators.

In addition to the aforementioned OTOC involving the qubit 1, we further investigate the OTOC with only the qubits 2 and 3. For the three systems, Figs. 9(d)-9(f) present the OTOC $F_{2,3}^{xx} = \langle \sigma_2^x(\tau)\sigma_3^x(0)\sigma_2^x(\tau)\sigma_3^x(0) \rangle$ and $F_{2,3}^{zz} = \langle \sigma_2^z(\tau)\sigma_3^z(0)\sigma_2^z(\tau)\sigma_3^z(0) \rangle$; permuting the site indices 2 and 3 in the above definitions yields similar results. As is the case in Figs. 9(a)-9(c), the OTOC for the free fermion and perturbed systems are semiquantitatively the same. Additionally, qualitative differences between these two systems and the quantum chaotic system emerge only in their convergence. Furthermore, in the free fermion system [Fig. 9(d)], $F_{2,3}^{xx}$ and $F_{2,3}^{zz}$ exhibit similar behavior, except for a sign difference in the asymptotic values. Consequently, the OTOC $F_{2,3}^{xx}$ and $F_{2,3}^{zz}$, as well as $F_{23,1}^{xx,z}$ and $F_{23,1}^{zz,z}$, are insufficient to distinguish the different types of information propagation channels, underscoring the significant advantage of the QRP.

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
