# Peer review of "Quantum reservoir probing: an inverse paradigm of quantum reservoir computing for exploring quantum many-body physics"

_SciPost Physics, doi:SciPost Phys. 18, 198 (2025)_

## Round 1 · Referee Report · Anonymous (Referee 1) · 2024-11-7

Strengths

  1. The authors introduced a new concept, quantum reservoir probing, which I think is interesting and worth exploring to deepen understandings of properties of quantum systems.
  2. Numerical simulations successfully demonstrate how well the authors’ proposal works.

Weaknesses

  1. This study lacks discussions on possible downsides of this framework.
  2. There is no theoretical guarantee on performance and efficiency.

Report

This paper introduces a new research question on the connection between properties of quantum systems and their information processing capabilities. To answer this question, the authors propose a new paradigm called quantum reservoir probing, which uses the framework of quantum reservoir computing to diagnose the property of quantum systems. The proof of concept is shown using numerical simulations for quantum Ising chain with transverse and longitudinal magnetic fields. The manuscript is well-written, but some concerns prevent me from recommending the manuscript for publication in the journal.

- In line 133, the authors mentioned that “Successful (unsuccessful) estimation indicates that the input information does (does not) influence the read-out operator”. However, does unsuccessful estimation always mean input information does not influence operator? The error could arise in the weight optimization. Does the error have an effect on the judge of the information propagation? Also, does a finite number of measurement shots affects the judge?
- Why do the authors consider 2-local observables at most? I agree that the increase of the number of operators does not necessarily mean the improvement of performance for quantum reservoir computing. On the other hand, I assume global observables could provide some valuable information for certain tasks. For instance, although it is a task with static quantum data, the phase recognition task could require information of global information. For quantum dynamical systems, is there no situation where the global information is needed? If it is not true, I think it is important to see if the QRP framework can efficiently capture the global information as well. It would be great if the authors could elaborate on it.
- Due to the definition of QRC, the result does not depend on the initial state of the quantum reservoir system. On the other hand, the result seems dependent on the initial state of the ancilla qubits used for input injection. How can we interpret this? Could it be possible to eliminate the effect of the dependence on the input-initial state?
- Why do the authors start with the ground state, despite the fact that the QRC is not influenced by the initial state. Is the washout time not enough to guarantee the independence on the initial state?
- As shown in the literature of reservoir computing, the range of input have an impact on the information capability. E.g., see a work by Kubota et al on the information processing capacity: https://journals.aps.org/prresearch/abstract/10.1103/PhysRevResearch.3.043135. Does it affect the performance of the QRP as well? If these factors including ones mentioned above matter, is it fair to say the QRP always tell the property of quantum systems? Could it be possible to provide certain theoretical guarantee?
- The short-term memory is used to see the information propagation in this work. Actually, there is a metric called the information processing capacity (IPC) in the context of reservoir computing that is used to see the profile of the reservoir’s ability to process the time-series data. Can it be used to see the nonlinear processing of the information through the quantum channels?
- The authors mentioned that an advantage of the framework is efficient operation compared to OTOC and TMI. I feel it boils down to the 2-localness of the observables considered for QRP, in contrast to these methods requiring the global information. In case the target property of the system is global, does the statement that the QRP is efficient still hold? In addition, does the result for QRP mean that observing local operator is enough to perform the task in the manuscript? Also, do the additional washout time or longer training and testing period lead to less efficiency of the proposal compared to other method?
- As for the applicability, how likely do we have the information of the input in practical situations? To perform QRP, we always need to have a supervised-learning-like setting. Thus, I wonder if it is likely to happen in practical settings. Moreover, in QRP, the dynamics should be expressed as Eq.(1). Can we extend this assumption to the case of unitary evolution, which could also be a common target property in quantum physics.
- Why do the authors consider the STM task as a function of the virtual time \tau, not k. In the original QRC, the STM objective function is a function of k. Thus, if we follow the concept of QRC straightforwardly, I think it makes sense to regard the objective function in the same way. It would be great if the authors could elaborate on the reason why the virtual time is introduced and it is the main parameter of the STM task. The way the virtual time is used is also different from the QRC perspective; QRC uses the virtual node to improve the expressivity. Therefore, I would recommend to note the difference in the manuscript.

Requested changes

  1. There is an inconsistency in reference; e.g., Initials are used for the first and family names in [5], while others use the initials for first names only.
  2. Some sentences are confusing; e.g., what does “general” mean in “Hereafter, we denote 〈O(ktin +τ)〉 for general k by 〈O(τ)” in line 162?

Recommendation

Ask for minor revision

  • validity: good
  • significance: ok
  • originality: high
  • clarity: good
  • formatting: reasonable
  • grammar: acceptable

Author:  Kaito Kobayashi  on 2024-12-28  [id 5069]

(in reply to Report 1 on 2024-11-07)

Please find the attached response.

Attachment:

reply.pdf

---

## Round 2 · Referee Report · Anonymous (Referee 1) · 2025-1-24

Report
・I asked in the previous round that “On the other hand, the result seems dependent on the initial state of the ancilla qubits used for input injection.”. I might be unclear and let me clarify again. My point was that the initial state with which the input-dependent state is created could be dependent on the performance. In the numerical experiments, the authors used the computational basis $|00\rangle$ and $|11\rangle$ to construct $|\psi_{in}(s^{k})\rangle$. That is, every time step, the input is injected on the $|00\rangle= (\frac{I+Z}{2})^{\otimes 2}$ or $|11\rangle= (\frac{I-Z}{2 })^{\otimes 2}$, meaning the way that the input comes into the system is always limited, i.e., through tensor products of (combinations of) $I$ and $Z$. Here $I$ is identity operator and $Z$ is Pauli $Z$. I know this is the conventional way of input-injection in QRC, but considering the fact that the types of input affect the performance, I assume this also have an effect on the precision for probing the property of quantum systems.
It would be nice if the authors clarify this.
・Related to the comment above, I would recommend the authors to examine the effect of hyperparameters such as types of input, evolution time, virtual time, initial state of the ancilla state and so on, because this would be an important point to see if the QRP can robustly produce the property of quantum systems regardless of practitioners' choice of hyperparameters. Then, it would be nice if you summarize this, e.g., in certain section or a paragraph in Discussion and conclusion. If these really affect the performance, I need to rethink the novelty of this work. (I mean, I would like to make sure if the method can perform the task well with only access to practically-reasonable prior knowledge on the system.)
Minor comment
I would recommend to double check typos: I found a typo, e.g., missing punctuation in a sentence “... estimate the information stored in O(τ) Employing...” in line 202, page 7.
Recommendation
Ask for minor revision

Author: Kaito Kobayashi on 2025-02-04 [id 5185]
(in reply to Report 1 on 2025-01-24)Please find the attached response.
Attachment:
reply.pdf

---

## Round 2 · Author Response

With this resubmission, we have attached a PDF containing our replies to the comments (reply.pdf). Please find it in the “Reports on this Submission” section of our “SciPost Submission Page.”
Sincerely,
Kaito Kobayashi and Yukitoshi Motome

---

## Round 2 · List of Changes

[1] 1st paragraph in Sec. 2.2: We rephrased a sentence to emphasize that the washout process ensures independence from the initial state.
[2] 1st paragraph in Sec. 2.2: We added sentences to further clarify the concept of virtual time in the QRP.
[3] 1st paragraph in Sec. 2.2: We revised a previously confusing sentence (line 162 in the prior manuscript).
Sec. 3
[4] 1st paragraph in Sec. 3.1: We added a description explaining that our input protocol preserves to the intrinsic symmetry of the system.
Sec. 4
[5] 3rd paragraph in Sec. 4: We included a sentence to emphasize the potential applicability of information processing capacity and non-local observables as promising avenues for future research.
Others
[6] We resolved the inconsistency in Refs. [5, 17].
[7] We added a new reference [71].
[8] For greater clarity, we revised the wording in the manuscript as necessary.

---

## Round 3 · Referee Report · Anonymous (Referee 1) · 2025-4-8

Report

I appreciate the authors’ revision. I am happy with the current form.

Recommendation

Publish (meets expectations and criteria for this Journal)

---

## Round 3 · Referee Report · Anonymous (Referee 2) · 2025-4-10

Strengths

  • Originality of the approach

Weaknesses

-Restricted analysis of parameter space -Unexplored role of memory effects - Lack of discussion about critical points

Report

The authors build upon quantum reservoir computing (QRC) paradigms, proposing an inverse approach that exploits QRC dynamics as a probing tool to detect properties of quantum many-body systems.

After outlining the general features and properties of QRC, the authors introduce a protocol that aims to reconstruct the value of a random input number via linear regression of dynamical observables, as usually done in reservoir computing.

The idea of the work is certainly interesting, however, there are several aspects that, in my opinion, need to be improved in order for the manuscript to be accepted for publication in SciPost Physics.

First of all, I see many qualitative observations (if the dynamics of systems in different phases are different, it is not surprising that the computational performance looks different). While the authors mention that such performance can be used as a metric, I fail to see it.

Furthermore, the protocol seems to rely on a systematic scan of different read-out operators, which is not necessarily an efficient method.

In most of the presented results, the system is employed as an extreme machine learning model—specifically, for zero-delay tasks where memory effects are irrelevant. However, the presence or absence of memory in the dynamics is a fundamental property that distinguishes ergodic and nonergodic phases in quantum systems. This critical aspect is entirely overlooked in the manuscript, despite its direct relevance to both quantum many-body physics and reservoir computing performance. A discussion of how memory effects—or their absence—manifest in the proposed protocol would significantly strengthen the work, particularly in clarifying whether the method probes purely instantaneous properties or can also capture time-dependent correlations in the quantum system.

The study focuses exclusively on two specific points in the parameter space—one associated with chaotic dynamics and the other with ballistic information propagation—both of which are well-characterized in prior literature. While this approach provides a useful starting point, it leaves several critical questions unanswered. Most notably:
Phase Boundaries and Criticality
Generality Across Parameter Space
Broader Implications for Critical Phenomena: If, as I suspect, the protocol performs well near critical points, this could open new avenues for data-driven detection of quantum phase transitions.

In summary, while the proposed inverse QRC approach is innovative and theoretically intriguing, the study in its current form requires substantial strengthening to meet the standards for publication in SciPost Physics.

Recommendation

Ask for major revision

  • validity: good
  • significance: high
  • originality: high
  • clarity: ok
  • formatting: excellent
  • grammar: excellent

Author:  Kaito Kobayashi  on 2025-04-22  [id 5394]

(in reply to Report 2 on 2025-04-10)

Please find the attached response.

Attachment:

reply.pdf

---

## Round 3 · Author Response

We extend our sincere appreciation to the editor for managing our manuscript. We are equally grateful to the reviewer for providing an excellent report that has greatly enhanced our work.
With this resubmission, we have included a PDF file (reply.pdf) with our detailed responses to the reviewer’s comments. Please find it in the “Reports on this Submission” section of our “SciPost Submission Page.”

Sincerely,
Kaito Kobayashi and Yukitoshi Motome

---

## Round 3 · List of Changes

Sec. 4
[1] 3rd paragraph in Sec. 4: We interchanged the original third and fourth sentences to enhance readability.
[2] 3rd paragraph in Sec. 4: We introduced additional sentences to highlight the design flexibility of QRP, which facilitates a wide range of potential applications. (Note: Although our reply.pdf stated that this revision related to the second paragraph, we repositioned it for better logical flow.)
Others
[3] We corrected typographical errors throughout the manuscript and refined the phrasing for improved clarity and precision.

---

## Round 4 · Referee Report · Anonymous (Referee 2) · 2025-5-19

Report

I appreciate the authors' efforts to address my criticisms and observations. While I don't agree with all the responses, especially regarding the generalizability of the results, I believe the manuscript is a solid piece of research and should be published. As I do not see any minor point that could be improved, I will give a positive recommendation for publication in SciPost Physics

Recommendation

Publish (meets expectations and criteria for this Journal)

---

## Round 4 · Author Response

We extend our sincere appreciation to the editor for managing our manuscript. We are equally grateful to the reviewers for their insightful comments, which have significantly improved our work.
Our detailed responses to the reviewers’ comments are provided in the attached PDF file (reply.pdf), available in the “Reports on this Submission” section of our SciPost Submission Page.

Sincerely,
Kaito Kobayashi and Yukitoshi Motome

---

## Round 4 · List of Changes

Sec. 2
[1] We added a new subsection Sec. 2.3, titled “General remarks on the QRP”.
[2] 1st paragraph in Sec. 2.3: We added a discussion clarifying our choice of zero‑delay tasks and highlighting how finite‑delay tasks can probe memory‑related phenomena, including ergodicity.
[3] 1st paragraph in Sec. 2.3: We added descriptions to emphasize the importance of the flexibility to choose read-out operators.
[4] 1st paragraph in Sec. 2.3: We added a sentence introducing the applicability of the QRP to quantitative analysis, such as perturbative effects.
Sec. 4
[5] 3rd paragraph in Sec. 4: We modified the discussion about the QRP configurations according to the addition of Sec. 2.3.
Others
[6] We refined the phrasing for improved clarity and precision.
[7] We added a reference [51].

---

## Editorial Decision

published